# Systemic Lupus Erythematosus and Lung Involvement: A Comprehensive Review

**DOI:** 10.3390/jcm11226714

**Published:** 2022-11-13

**Authors:** Jae Il Shin, Keum Hwa Lee, Seoyeon Park, Jae Won Yang, Hyung Ju Kim, Kwanhyuk Song, Seungyeon Lee, Hyeyoung Na, Yong Jun Jang, Ju Yun Nam, Soojin Kim, Chaehyun Lee, Chanhee Hong, Chohwan Kim, Minhyuk Kim, Uichang Choi, Jaeho Seo, Hyunsoo Jin, BoMi Yi, Se Jin Jeong, Yeon Ook Sheok, Haedong Kim, Sangmin Lee, Sangwon Lee, Young Soo Jeong, Se Jin Park, Ji Hong Kim, Andreas Kronbichler

**Affiliations:** 1Department of Pediatrics, Yonsei University College of Medicine, Seoul 03722, Republic of Korea; 2Yonsei University College of Medicine, Seoul 03722, Republic of Korea; 3Department of Nephrology, Yonsei University Wonju College of Medicine, Wonju 26426, Republic of Korea; 4Department of Pediatrics, Eulji University School of Medicine, Daejeon 34824, Republic of Korea; 5Department of Pediatrics, Gangnam Severance Hospital, Yonsei University College of Medicine, Seoul 26426, Republic of Korea; 6Department of Medicine, University of Cambridge, Cambridge CB2 0QQ, UK

**Keywords:** systemic lupus erythematosus, autoimmunity, pleuropulmonary, lung, review

## Abstract

Systemic lupus erythematosus (SLE) is a complex autoimmune disease with multiorgan manifestations, including pleuropulmonary involvement (20–90%). The precise mechanism of pleuropulmonary involvement in SLE is not well-understood; however, systemic type 1 interferons, circulating immune complexes, and neutrophils seem to play essential roles. There are eight types of pleuropulmonary involvement: lupus pleuritis, pleural effusion, acute lupus pneumonitis, shrinking lung syndrome, interstitial lung disease, diffuse alveolar hemorrhage (DAH), pulmonary arterial hypertension, and pulmonary embolism. DAH has a high mortality rate (68–75%). The diagnostic tools for pleuropulmonary involvement in SLE include chest X-ray (CXR), computed tomography (CT), pulmonary function tests (PFT), bronchoalveolar lavage, biopsy, technetium-99m hexamethylprophylene amine oxime perfusion scan, and (18)F-fluorodeoxyglucose positron emission tomography. An approach for detecting pleuropulmonary involvement in SLE includes high-resolution CT, CXR, and PFT. Little is known about specific therapies for pleuropulmonary involvement in SLE. However, immunosuppressive therapies such as corticosteroids and cyclophosphamide are generally used. Rituximab has also been successfully used in three of the eight pleuropulmonary involvement forms: lupus pleuritis, acute lupus pneumonitis, and shrinking lung syndrome. Pleuropulmonary manifestations are part of the clinical criteria for SLE diagnosis. However, no review article has focused on the involvement of pleuropulmonary disease in SLE. Therefore, this article summarizes the literature on the epidemiology, pathogenesis, diagnosis, and management of pleuropulmonary involvement in SLE.

## 1. Introduction

Systemic lupus erythematosus (SLE) is a complex autoimmune disease with potential multiorgan involvement, including pulmonary involvement, arthritis, photosensitive rashes, glomerulonephritis, and cytopenia [1,2]. SLE is an extremely heterogeneous disease, and its pathophysiology remains unknown [3]. The lack of pathognomonic molecular markers or nonspecific constitutional symptoms delays the early diagnosis and treatment of SLE [4]. This complex clinical feature and unknown pathophysiology make the diagnosis of SLE difficult. In 2019, the European League Against Rheumatism (EULAR)/American College of Rheumatology (ACR) developed new SLE classification criteria, including one obligatory entry criterion (positive antinuclear antibody (ANA)) followed by additional weighted criteria grouped into seven clinical (constitutional, hematologic, neuropsychiatric, mucocutaneous, serosal, musculoskeletal, and renal) and three immunologic (antiphospholipid antibodies, complement proteins, and SLE-specific antibodies) domains, weighted from 2 to 10 [5].

Pleural effusion is included in the current EULAR/ACR criteria. Pleuropulmonary manifestations, including pleural effusion, are part of the clinical criteria and are highly prevalent in SLE [6]. Many (50–70%) patients with SLE experience pulmonary complications ranging from subclinical pleural effusion to life-threatening alveolar hemorrhage [6], and 4–5% have pulmonary manifestations as presenting symptoms [7]. Pulmonary manifestations in children occur less frequently than in adults; however, they could be significant and lethal SLE complications [8,9]. Various studies have reported that patients with SLE may have symptoms of subtle or absent pulmonary dysfunction, suggesting a subclinical disease [10,11,12,13]. One multiethnic United States cohort study (LUMINA XLVIII) [14] reported that 7.6% and 11.6% of patients had permanent lung damage 5 and 10 years after SLE diagnosis, respectively. Age, pneumonitis, and anti-ribonucleoprotein (anti-RNP) antibodies are positively correlated with the early development of permanent lung disease [14]. In another retrospective (RELESSER) cohort study [15], patients with pleuropulmonary manifestations had significantly lower survival rates than those without pleuropulmonary manifestations (82.2% vs. 95.6%, *p* = 0.030). Thus, SLE-related pleuropulmonary manifestations are potentially life-threatening in children and adults with SLE. An in-depth understanding of SLE-related pleuropulmonary manifestations will help facilitate early diagnosis and prevent the disease from progressing to a serious condition. Therefore, this review aimed to describe the epidemiology, pathophysiology, diagnosis, and management of SLE-related lung diseases, including infection and drug-induced lung injury.

## 2. Epidemiology

### 2.1. General Incidence

The term “lupus erythematosus” was initially used by physicians to describe skin lesions in the 19th century [16]. However, the systemic characteristics, the extent of organ involvement, and clinical heterogeneity of the disease caused by an abnormal autoimmune response were not realized until 100 years later [17]. SLE may affect vital organs such as the brain, blood, kidneys, and predominantly occurs in women of childbearing age [16].

The prevalence of SLE varies according to the sampling and recruitment methods used in studies and ranges from 20 to 150 cases per 100,000 persons in the United States [16,18]. The incidence of the disease nearly tripled by the end of the 20th century owing to the improved detection of mild SLE [19].

Geographic and racial distributions appear to affect the prevalence and severity of SLE in clinical and laboratory settings. People with African or Asian backgrounds have approximately two to three times higher incidence and prevalence rates in the United States and the United Kingdom than other ethnicities [20,21,22,23,24,25]. In more detail, Asians, African Americans, Caribbeans, and Hispanic Americans have a higher prevalence of SLE than Caucasians [18,26,27]. Asian and African descents in European countries also show similar results, while Africa reports the lowest prevalence of SLE [18].

Sex is the most important risk factor for SLE, as 90% of patients with SLE are women in most studies [18]. Hormones contribute to the onset of SLE through an unknown mechanism [17]. Cluster of differentiation 40 is located on chromosome X, one of the genes responsible for the pathogenesis of SLE [16]. Since sex hormones have minimal effects on children, the women-to-men ratio of patients with SLE is 3:1, while it ranges from 7 to 15:1 in adults, especially in women of childbearing age [20]. Postmenopausal women have lower grades of sex hormones; hence, the women-to-men ratio is decreased to 8:1 in older adults [28].

SLE can occur throughout life, and approximately 20% of cases are diagnosed within the first 20 years. It is quite rare before the age of 5 years; yet, its prevalence increases after the first decade [29,30,31,32]. The prevalence and incidence rates of SLE in adulthood are considerably higher [16]. In patients with late-onset SLE, the women-to-men ratio and disease activity are lower while organ damage accrual is greater, leading to higher mortality [33].

### 2.2. Incidence of Pleural Involvement in SLE

Respiratory involvement in SLE is frequent, especially in men, as nearly 50% of patients with SLE have lung involvement during the course of their disease, and 4–5% present with pulmonary symptoms [7,34]. The most favorable thoracic localization involves the pleura and ranges from an asymptomatic manifestation to pleuritic pain, present in 45–60% of patients. The latter can also be accompanied by radiographically undetectable chest effusion [34]. The pathologic evidence of pleural fibrosis or pleuritis is 50–93%, and pleural effusion in chest radiologic studies is 16–50% [34].

### 2.3. Incidence of Pulmonary Parenchyma Involvement in SLE

The clinical manifestations of lung parenchyma involvement in SLE can be acute or chronic [34]. Some (2–9%) of the cases present with acute lupus pneumonitis (ALP) and 3–9% interstitial lung disease (ILD) in previously undiagnosed SLE [34,35,36]. Diffuse alveolar hemorrhage (DAH) is a rare but critical manifestation, with a mortality rate of 50–90%, a prevalence of 0.5–5.7%, and a women-to-men ratio of 6:1 [37,38,39,40,41,42,43,44,45,46,47].

## 3. Pathogenesis

SLE is an autoimmune disease characterized by the production of antibodies against cell nucleus components. The primary pathological findings in patients with SLE include inflammation, vasculitis, immune complex deposition, and vasculopathy. Multiple genes confer susceptibility to disease development. The interaction of sex, hormonal milieu, the hypothalamic–pituitary–adrenal axis, and defective immune regulation, such as the clearance of apoptotic cells and immune complexes, modify this susceptibility. The loss of immune tolerance, increased antigenic load, excess T-cell activities, defective B-cell suppression, and shifting of Th1 to Th2 immune response causes cytokine imbalance, B-cell hyperactivity, and the production of pathogenic autoantibodies [48]. 

Based on this perspective, inflammation and immune response dysregulation are important drivers of lung pathology in autoimmune diseases [14]. The precise mechanism of lung involvement in SLE is unknown; however, several features are associated with SLE, such as elevated systemic type 1 interferon (IFN) levels, circulating immune complexes (IC), and neutrophils. These factors and other unidentified mediators seem to play essential roles in driving lung inflammation and, ultimately, fibrosis and tissue damage [49]. 

Patients with SLE and lung involvement have increased levels of systemic proinflammatory cytokines [50,51]. In patients with lung involvement, the levels of proinflammatory cytokines such as IFN-γ, tumor necrosis factor α (TNF-α), interleukin-6 (IL-6), interleukin 8 (IL-8), and interleukin-12 are two or three fold increased in comparison to patients without lung involvement, and the IL-8/IL-10 ratio was 3 fold increased the level in restrictive lung disease [50,51]. While the role of inflammation in the progression of pulmonary fibrosis is unclear, cellular inflammation, particularly in the early stages of the disease, is consistent with the pathological findings [50,51]. As an effect of initial inflammatory insult (injury, infection, or antibody deposition), damaged or activated epithelial or endothelial cells release proinflammatory cytokines or chemokines (such as TNF-a, IL-1, and IL-8), resulting in the attraction and homing of neutrophils, followed by monocytes, macrophages, and T- and B-lymphocytes [50,51]. In SLE, neutrophils further play a contributing role, because they release DNA and histones in a process called NETosis, revealing autoantigens and self-DNA to further exacerbate inflammatory responses [52]. Type 1 IFNs also play an essential role in driving neutrophil NETosis, autoantibody production, and breaking the immune tolerance in the lungs [49]. This reciprocal interaction between the initiating factors (IFNs, autoantibodies, immune complexes, infectious insult, or injury) and downstream responses (complement activation, neutrophil accumulation, and activation) likely plays an essential role in driving SLE-associated lung involvement [49]. Macrophage activation syndrome (MAS) has been described in several case reports of patients with lupus; however, its relationship is unclear and is rarely the first presentation of SLE [53,54,55]. In juvenile SLE, some cases of MAS that developed in the initial stage should be treated with immunosuppressants [56].

In recent decades, many studies have been published on the pathogenesis of SLE-associated lung involvement (Figure 1). Table 1 summarizes the major studies on the pathogenesis of SLE-associated lung involvement.

## 4. Diagnosis

### 4.1. Chest X-ray (CXR)

The CXR findings vary for various diseases. Pleural effusion was observed in patients with pleural involvement [35]. In ALP, the CXR frequently demonstrates nonspecific pleural effusions and acute development of consolidation in one or more areas (usually basal and bilateral) [35]. Regarding chronic ILD, although patients usually have clinical symptoms such as dyspnea on exertion, pleuritic pain, dry cough, and rales, the results may be normal or show irregular opacities in the early stages [61]. However, in later stages, it shows diffuse or bi-basal infiltration, pleural disease, honeycomb (coarse reticular opacities), and diminished lung volume [35]. Regarding DAH, the radiograph might show diffuse hemorrhage, a diffuse infiltrative opacification pattern with a slight predilection towards the mid zones with some apical sparing [35]. In shrinking lung syndrome (SLS), a rare and less-known SLE complication that shows diaphragmatic dysfunction and reduced diffusing capacity for carbon monoxide (DLco), a typical finding is elevated hemidiaphragm and reduced lung volume [62,63,64]. Interstitial infiltrates, pleural thickening, and basal atelectasis can also be found, but the CXR results are normal in some cases [65,66].

As described above, previous studies on CXR findings in patients with SLE with lung involvement were descriptive and did not report the prevalence of each manifestation. However, in symptomatic patients, CXR results usually correlate with clinical symptoms. As in the early stages of chronic ILD and SLS, images can demonstrate normal findings [35]. Therefore, chest radiography is the most basic imaging tool for assessing lung involvement in SLE. However, its sensitivity for screening lung involvement in SLE is low, suggesting the need for another diagnostic tool.

### 4.2. Computerized Tomography (CT)

Early diagnosis of pulmonary involvement in patients with SLE can be achieved using high-resolution CT (HRCT). In adults, 70% of patients with SLE showed abnormal findings on HRCT, such as thickened interlobular septa (44%), parenchymal bands (44%), subpleural bands (21%), bronchiectasis (21%), and bronchial wall thickening (21%) [67]. CT imaging in symptomatic children with SLE also shows ground-glass opacity (GGO) (33%), mosaic appearance (27%), pleural irregularity (27%), and bronchial wall thickening (27%) [68]. Some children with SLE without clinical manifestations of pulmonary diseases also showed HRCT findings, such as GGO, consolidation, and pleural irregularity [68]. Therefore, HRCT helps to detect early-stage pulmonary involvement in symptomatic and asymptomatic patients with SLE.

A study from China showed GGO in ten (55.6%) patients, interlobular septal thickening in nine (50.0%), bilateral diffuse infiltrates in nine (50.0%), pleurisy/pleural effusion in eight (44.4%), fibrotic streaks in six (33.3%), and patchy infiltrates in five (27.8%) (Figure 1). Most abnormalities were bilaterally distributed (94.4%) in the lower lobes (82.3%) or subpleural regions (61.1%) (Figure 2) [69].

Patients with pulmonary fibrosis, such as ILD, had GGO, and the extent of the disease, according to the CT, correlates well with the spirometric values, such as forced expiratory volume in the first forced expiratory volume/forced vital capacity (FEV1/FVC) and DLco [70]. The results suggest that CT imaging is helpful for patients with SLE with pulmonary involvement owing to its relatively precise expectation of pulmonary function. Furthermore, some HRCT findings of patients with ILD correlated with open lung biopsy [67]. In some cases, patients with SLE (21%) were diagnosed with ILD using HRCT while there were no significant changes in the CXR and pulmonary function test (PFT) results [67]. Therefore, abnormal HRCT changes allow the diagnosis of pulmonary involvement which is undetectable by other diagnostic tools.

### 4.3. Pulmonary Function Test

PFT can be an effective diagnostic tool for patients with SLE and subclinical pulmonary disease. PFT allows the early detection of patients with SLE with pulmonary involvement, because they show low DLco without any respiratory clinical manifestations [71]. Furthermore, some patients with SLE only showed a reduction in DLco without any abnormalities in imaging studies [71]. Among the measured spirometric values, such as the FVC, FEV1/FVC ratio, total lung capacity (TLC), and functional residual capacity (FRC), the most prevalent PFT impairment was DLco [71]. The broad availability of PFT makes it the preferred method; however, it has some limitations, such as abnormal results in patients with obstruction, restriction, or abnormal diffusing capacity [72]. Therefore, some patients with SLE and pulmonary involvement may not be detected using PFT. In such cases, other imaging tools or the six-minute walk test, which evaluates the total distance walked, oxygen desaturation, and the patient-reported Borg Dyspnea Scale after walking back and forth down a 30-m hallway, can help to detect pulmonary diseases [72].

### 4.4. Biopsy

In patients with pleural involvement, fibrosis, lymphocytic and plasma cell infiltration, and fibrinous pleuritis were detected, and anti-IgG, anti-IgM, and anti-C3 nuclear patterns were seen in the immunofluorescence tests [73]. ALP shows alveolar wall damage, necrosis, inflammatory cell infiltration, edema, hemorrhage, and hyaline membranes [73]. The predominant finding in DAH is bland hemorrhage, capillaritis, immune complex deposition, and neutrophils in the alveolar walls at lower rates [45,47].

However, a biopsy is only used when the diagnosis is uncertain and other findings are nonspecific [73]. Postoperative complications, morbidity, and mortality are factors limiting the use of lung biopsy [74].

### 4.5. Bronchoalveolar Lavage (BAL)

BAL is mainly used to exclude other causes of respiratory compromise. It can be used to rule out infection as an underlying cause of ALP, and BAL shows an increase in cellularity because of the activation of polymorphonuclear cells in such cases [51]. One study showed the predominance of lymphocytes and neutrophils in ILD; however, SLE without pulmonary symptoms showed a predominance of macrophages [75]. In patients with nonspecific symptoms, BAL is useful to confirm DAH, which is defined as ≥20% of hemosiderin-laden macrophages in bronchoalveolar lavage fluid (BALF) [76,77,78].

### 4.6. FDG-PET

In 2020, a study showed that a high FDG uptake in pulmonary arterial hypertension (PAH) in patients with SLE correlated with SLE disease activity and the immune/inflammatory status (C3 and C4 levels) [79]. Thus, FDG-PET may assess intrapulmonary disease activity markers in patients with SLE-PAH [79].

### 4.7. Dynamic Magnetic Resonance Imaging (MRI)

An MRI can confirm pleuritis in SLE-associated shrinking lung syndrome. Dynamic MRI sequences on forced voluntary ventilation can confirm the physiological movements of the diaphragms and all auxiliary respiratory muscles [80,81]. The evaluation of diaphragm dome motion using a dynamic contrast-enhanced lung MRI might be a useful second-line tool to reinforce clinical suspicion in cases of diagnostic difficulty. It is not as widely used as other imaging studies; however, a dynamic MRI can sufficiently diagnose this disease.

### 4.8. Summary

The diagnosis of pulmonary involvement is made by using different imaging tools [67]. However, using a single tool to define pulmonary involvement is insufficient. For example, there is little evidence for the use of PFT as a screening method, as lung involvement is uncommon and is mostly acute [82]. Reductions in DLco and the maximum voluntary ventilation are generally not helpful to evaluate if pulmonary involvement is present [82]. Therefore, pulmonary screening in SLE should include chest CT and PFTs, which are mildly invasive, easily available, and can sometimes detect early manifestations before symptoms occur. Additional studies, such as a dynamic MRI, should be performed for lung-involved diseases with poor prognoses (Table 2). 

## 5. Types of Pulmonary Involvement

SLE can cause various pulmonary diseases, such as pleural effusion/pleuritis, shrinking lung syndrome, acute pneumonitis, DAH, chronic ILD, and pulmonary hypertension [35]. Pleuritis is the most common intrathoracic disease in SLE, which presents with chest pain, cough, dyspnea, and pleural effusion [34,83]. ALP and DAH are acute pulmonary diseases resulting from damage to the alveolar–capillary unit [83], while ILD is a chronic pulmonary disease that causes pulmonary fibrosis [83]. We summarize the clinical manifestation, treatment, sequelae, and outcomes according to the type (Table 3).

### 5.1. Lupus Pleuritis

#### 5.1.1. Clinical Manifestation

Pleuritis, also known as pleurisy, is an inflammation of the pleural tissue that causes sharp chest pain (pleuritic pain), which worsens during breathing. Pleuritis with or without pleural effusion is the most common feature of acute pulmonary involvement in SLE [84]. It is characterized by chest pain, dyspnea, shortness of breath, cough, and possibly fever. Pleuritis is often difficult to diagnose without pleural effusion on a chest X-ray. Pleural friction rubbing may facilitate the diagnosis of pleuritis [84]. A pleural US or HRCT combined with serological markers for SLE activity can confirm pleuritis in vague cases [85].

Pleural effusion in SLE is often bilateral (50%), and the amount is usually small to moderate. The pleural fluid appearance varies from clear serous to serosanguineous and bloody [84]. It is an exudate characterized by elevated lactate dehydrogenase and a white blood cell count of 3000–5000 cells/mL, with a predominance of neutrophils or mononuclear cells [83]. Hemopneumothorax has been described in patients with lupus pleuritis; however, spontaneous pneumothorax is uncommon [86]. Fibrothorax is a rare complication that can lead to dyspnea and shortness of breath by preventing lung expansion [87].

#### 5.1.2. Treatment

Treatment options for lupus pleuritis differ depending on the severity of symptoms. Mild pleuritis can be treated with NSAIDs [88]. If a patient has difficulty enduring the gastrointestinal side effects of NSAIDs, the dosage may be adjusted [88], and NSAIDs should be avoided in patients with impaired kidney function. Antimalarials such as hydroxychloroquine may be added in some cases [89].

Systemic corticosteroids are the drugs of choice for patients with severe lupus pleuritis. Hunder et al. found that five out of six patients with lupus pleuritis treated with corticosteroids experienced a rapid disappearance of pleural effusions [90]. Winslow et al. reported that effusions in 10 out of 11 patients subsided quickly after treatment [91]. If the volume of pleural fluid is large, aspiration may be required to alleviate respiratory difficulties [87].

Cyclophosphamide should be considered for patients who do not respond to corticosteroids or have concomitant kidney involvement [88]. Other options include mycophenolate mofetil (MMF) and rituximab. Kariem et al. showed that MMF lowered the extrarenal and renal symptoms in patients with SLE who did not respond to conventional therapies [92]. In another study by Ng et al., two out of seven patients with lupus pleuritis experienced improvement after depleting B cells with rituximab [93].

Intravenous immunoglobulin (IVIg) is not the mainstay treatment for lupus pleuritis; however, Meissner et al. reported that 1 month of IVIg therapy effectively decreased pleural effusion in patients with SLE and severe pleuritis [94]. Sherer et al. also reported that a combined therapy of IVIg and cyclosporin alleviated massive pleural effusion [95]. These reports suggest the utility of IVIg in patients with severe lupus; however, further studies are needed to confirm its clinical effectiveness.

A pleurectomy may be needed in rare cases of patients with refractory pleural effusion who do not respond to immunosuppressive drugs [96]. Sichan et al. successfully treated such patients by performing open surgical pleurectomy [96].

#### 5.1.3. Sequelae and Outcomes

Pleural diseases, especially lupus pleuritis, cause severe lung restriction and fibrothorax in chronic and refractory pleuritis [87]. Patients with acute pleurisy normally present with severe chest pain, dyspnea, fever, cough, and friction rub [97]. With pleural effusion and pleuritis, some patients were initially not diagnosed with SLE but with other pulmonary diseases. In one study, out of 14 patients diagnosed with lupus pleuritis, only five were diagnosed with SLE on admission. The rest were initially diagnosed with pulmonary embolism, congestive heart failure, and other pulmonary manifestations [97]. Lupus pleuritis was finally diagnosed based on a pleural fluid analysis and clinical presentation, indicating an SLE diagnosis. Since lupus pleuritis was misdiagnosed and inadequate treatment was provided, some patients developed rare complications, such as fibrothorax. Fibrothorax is a state of progressive pleural fibrosis that leads to limited lung expansion and dyspnea [87]. In one study, pleural stripping, also known as decortication, effectively treated thickened visceral pleura [87]. Lupus pleuritis has good treatment outcomes. In a study of 119 patients involving 127 pleuritis/pleural effusion episodes, 93% of the patients responded to corticosteroids, and out of these, 94% completely responded to the initial treatment. Relapse only occurred in approximately 13% of patients, and none progressed to fibrotic disease [98]. Lupus pleuritis is the most common pulmonary complication of SLE; however, its prognosis and treatment response are good.

### 5.2. Pleural Effusion

#### 5.2.1. Clinical Manifestation

According to the EULAR/ACR_2019 criteria, the only criterion that is related to lung involvement is pleural effusion, and the definition of this criterion is evidenced by imaging techniques: ultrasound, X-ray, CT scan, and MRI [5]. Pleural effusion in SLE may be caused by autoimmune pleuritis, but it cannot be easily distinguished from other causes, such as infections, heart disease, and tuberculosis [6]. Therefore, more detailed criteria for pleural effusion should be established.

#### 5.2.2. Treatment

Usually, SLE-associated pleural effusion responds quickly to corticosteroids [99]. However, persistent steroid-resistant pleural effusion can occur in some cases. Various methods have been used to treat these cases. Some cases of SLE-associated steroid-resistant pleural effusion have been successfully treated with tetracycline pleurodesis [100,101]. A case of SLE-associated refractory massive pleural effusion was successfully treated with pleurectomy [102].

#### 5.2.3. Sequelae and Outcomes

No study with a statistically meaningful sample size was found regarding the sequelae and outcomes of SLE-associated pleural effusion.

### 5.3. Acute Lupus Pneumonitis (ALP)

#### 5.3.1. Clinical Manifestation

ALP is an uncommon SLE manifestation [103]. The clinical presentation of ALP is similar to that of acute interstitial pneumonia, characterized by the acute onset of fever, cough, and dyspnea [104]. Sometimes, hemoptysis is accompanied by tachypnea, tachycardia, inspiratory crackles, and hypoxemia during the physical examination. Half of the patients with ALP experience pneumonitis as the initial SLE manifestation [104]. Diffuse alveolar damage, alveolar edema, hyaline membrane formation, and mononuclear cell infiltration can be found during histopathological examination. Immunoglobulin and complement deposition may be present in capillary walls [105,106]. Alveolar hemorrhage may also be noted, whereas vasculitis is uncommon [104]. Chest X-rays reveal diffuse or patchy opacities, predominantly in the lower lung zones. A differential diagnosis is needed to rule out infection, organizing pneumonia, pulmonary embolism, drug toxicity, DAH, heart failure, and malignancy [106].

#### 5.3.2. Treatment

Empirical broad-spectrum antibiotic therapy should be administered without delay because of the high incidence of infection [107]. After infectious etiologies are excluded, aggressive immunosuppressive therapy must be initiated. High doses of intravenous methylprednisolone, oral corticosteroids, and intravenous cyclophosphamide can be considered [91]. In addition, successful treatment with rituximab has been reported [108]. Plasma exchange and intravenous immunoglobulins have been used in refractory cases [109].

#### 5.3.3. Sequelae and Outcomes

Acute lupus pneumonitis is a complication of SLE that occurs in 1–12% of patients. It is a life-threatening condition characterized by nonspecific alveolar damage, necrosis, inflammatory cell infiltration, and edema [51]. Studies have not been reported since 1975, indicating that, out of 12 cases with acute lupus pneumonitis, six patients survived, and most had residual interstitial infiltrates that led to chronic interstitial pneumonitis [104]. Specific studies related to acute lupus pneumonitis have not been conducted; however, the current treatment with ventilators and plasma exchange has improved its prognosis. Patients are at risk of persistent pulmonary function abnormalities and restrictive lung disease in cases of infection [51]. The patient’s clinical presentation is similar to that of acute interstitial pneumonia. No specific treatment is available; however, most patients are treated with broad-spectrum antibiotics.

### 5.4. Diffuse Alveolar Hemorrhage

#### 5.4.1. Clinical Manifestation

DAH varies in its clinical intensity, ranging from mild to severe, potentially life-threatening pulmonary hemorrhage. Patients experiencing DAH present with dyspnea, cough, fever, blood-stained sputum, and sometimes hemoptysis, with symptoms developing rapidly within a few hours or days [78]. Typical conditions include a drop in the hemoglobin level and diffuse lung infiltrates visible on chest X-rays or high-resolution chest CT [78]. Infections can be associated with DAH and should be ruled out as a differential diagnosis [36]. BAL is useful for confirming DAH when ≥ 20% of hemosiderin-laden macrophages in BALF are present [78].

#### 5.4.2. Treatment

There are no specific treatment guidelines for DAH; however, high-dose methylprednisolone is often used [110]. Cyclophosphamide is another option, but its effectiveness is controversial. Zamora et al. found that cyclophosphamide caused increased mortality [38], which might have been confounded by an indication bias. In other studies, plasma exchange was used in combination with cyclophosphamide or methylprednisolone; however, its effectiveness as a monotherapy has not been established [77].

#### 5.4.3. Sequelae and Outcomes

The mortality rate of patients with SLE and pulmonary hemorrhage is very high. Evidence suggests a mortality rate of 68–75% [111,112] during follow-up. The latter study showed that the same proportion (75%) required mechanical ventilation [112]. Mortality is related to several factors. First, patients who experience severe dyspnea and require mechanical ventilation are at a higher risk of mortality [111,112]. This results from severe dyspnea associated with severe lung failure, and patients with long-term mechanical ventilation have a higher risk of ventilator-associated pneumonia [112]. Second, kidney failure in patients with pulmonary hemorrhage also contributes to poor outcomes [42]. Third, patients with neuropsychiatric complications are at a high risk of mortality [113]. Other factors such as thrombocytopenia, infections, age, and multi-organ failure scores also cause higher mortality [111].

However, recent studies have shown that the early and aggressive treatment of patients presenting with pulmonary hemorrhage can increase their survival rates [114]. This includes management with high-dose steroids, IVIG, and cyclophosphamide [76]. Infections occur concomitantly in approximately 57% of patients with pulmonary hemorrhage; therefore, managing these patients with the appropriate antibiotics is important to lower the mortality rate [115]. In addition, alternative and experimental therapeutic approaches, such as umbilical cord-derived mesenchymal stem cell transplantation, have been shown to improve outcomes [116,117].

### 5.5. Interstitial Lung Disease (ILD)

#### 5.5.1. Clinical Manifestation

The true prevalence of SLE-associated interstitial lung disease (SLE-ILD) is unknown; however, a prevalence rate of 3–9% has been reported [103,107,118]. Nonspecific interstitial pneumonia (NSIP), organizing pneumonia, lymphocytic interstitial pneumonia (LIP), follicular bronchiolitis, nodular lymphoid hyperplasia, and usual interstitial pneumonia are associated with SLE. Among these, NSIP is the most common [119]. Patients with SLE-ILD typically show an insidious onset of chronic nonproductive cough, dyspnea, decreased exercise tolerance, and basilar crackles on physical examination, but some patients are asymptomatic [120]. By identifying these clinical features, a SLE-ILD diagnosis is made on a clinical basis and by HR CT confirming ILD; while excluding other causes, such as infection, drug toxicity, and heart failure [119]. The serological markers for SLE did not show a good correlation with the development of ILD, but they helped confirm the SLE diagnosis [107].

#### 5.5.2. Treatment

SLE-associated ILD treatment is mainly based on expert opinion [121]. Recent treatment algorithms suggest induction therapy with corticosteroids alone or with cyclophosphamide or MMF and maintenance therapy with azathioprine or MMF.

#### 5.5.3. Sequelae and Outcomes

In a retrospective, a multicenter study from Japan, including 55 patients with SLE-related interstitial pneumonia, the overall 5-year survival rate was 85.3% [122]. Fifty-one out of fifty-five patients were treated for interstitial pneumonia, all of whom had received corticosteroids, and twenty-two out of these received immunosuppressive medications. Among the prognostic factors, smoking, thrombocytopenia, and the extent of lung fibrosis (extent scores of 2 or 3) on HRCT predicted worse outcomes [122].

In lupus-associated pneumonitis, another type of ILD, progression to respiratory failure is common and results in high mortality [111]. ALP may develop as part of a generalized SLE flare, typically in patients with multisystem involvement, including nephritis, arthritis, pericarditis, and pleuritis. Fulminant ALP levels may also occur during pregnancy. Patients with ALP may have an acute onset of fever, dyspnea, and cough with scanty expectoration, hemoptysis, and pleuritic chest pain [111]. The symptoms include cyanosis and bi-basilar rales [111]. Furthermore, the arterial blood gas analysis results showed hypoxia in almost all the patients. DAH can occur concomitantly, although DAH may also occur in other SLE-related entities, including antiphospholipid syndrome, and distinguishing between these entities can be challenging [123]. In addition, it is difficult to accurately estimate the exact incidence and outcome statistics of SLE-associated ILD, because many patients with lupus and transient pleural diseases do not show symptoms, and a diagnosis is often made late in the disease course.

### 5.6. Shrinking Lung Syndrome

#### 5.6.1. Clinical Manifestation

Shrinking lung syndrome (SLS) was first described by Hoffbrand and Beck in 1965 [124]. SLS is a rare SLE complication. Its estimated prevalence is <1% [65,125]. The appearance of SLS as the primary respiratory SLE symptom is very rare [126,127]. Few cases show that SLS was diagnosed at the time of SLE diagnosis. SLS is usually discovered after SLE diagnosis (the mean delay between SLE and SLS diagnosis is approximately 67 months) [66]. The sex ratio showed a higher prevalence in women [66]. All patients with SLS complained of respiratory symptoms such as dyspnea, coughing, and pleuritic chest pain [66,124]. SLS is characterized by a progressive decrease in lung volume on PFT and no evidence of interstitial disease or significant pleural disease on the chest CT [65,127].

#### 5.6.2. Treatment

There are no clinical guidelines for SLS owing to its rare incidence. However, many cases of SLS associated with SLE have been treated with glucocorticosteroids monotherapy or in addition with other immunosuppressive agents, such as cyclophosphamide, azathioprine, MTX, and mycophenolate mofetil [66,81,128]. Theophylline and beta-agonists can help with diaphragmatic weakness [129,130,131]. Rituximab has also been used successfully as a monotherapy or with cyclophosphamide and beta-agonists, especially in steroid-refractory cases [81,132,133]. Other drugs have been used successfully after steroid failure; however, in a retrospective study by Robles-Perez et al., 6/18 patients reported adverse effects of rituximab [134]. Belimumab is approved for non-renal SLE, and a case report showed improvement in SLS symptoms; however, further studies are needed [135].

#### 5.6.3. Sequelae and Outcomes

The etiology of SLS has been explained by the relationship between diaphragmatic weakness resulting from inflammatory myopathy and phrenic neuropathy. Elevation of the diaphragm leads to dyspnea and reduced lung volumes. Treatment with corticosteroids and immunosuppressive therapy showed good response rate. However, in some patients, diaphragmatic weakness persists after treatment and requires surgical treatment [136]. Most patients exhibit acute symptomatic improvement and reversible pulmonary function. In a study of 55 patients, 18% had no functional sequelae, the rest showed significant clinical improvement, and none required long-term oxygen therapy [81]. Most patients are usually healed from SLS; however, they must be aware of the associated complications, such as pneumonia, that can impact further prescription of immunosuppression and thus efficacy.

### 5.7. Pulmonary Arterial Hypertension

#### 5.7.1. Clinical Manifestation

PAH is not included in any pre-existing SLE diagnostic criteria; however, it is a rare complication with increasing recognition in patients with SLE. PAH must be accurately diagnosed and promptly treated, as it may lead to irreversible changes in the right ventricle or lung capillaries, causing high mortality in patients with PAH [137]. In large cohort studies, approximately 2–5% of patients with SLE were diagnosed with PAH [138,139]. The delay period between the diagnosis of SLE and PAH was approximately 3–5 years [138,140]. The 5-year survival rate was 70–80% [138,140].

Several efforts have been made to elucidate the clinical and laboratory risk factors and predictors of PAH in patients with SLE. Systemic hypertension, high fibrinogen levels, and the presence of serositis and thrombocytopenia are clinical risk factors, indicating that SLE complications in other organ systems may contribute to the pulmonary manifestations of SLE [138,139]. Scleroderma-like nailfold capillary patterns and the Raynaud phenomenon are also important risk factors for PAH in patients with SLE, in line with the well-established connection between scleroderma and PAH [141,142]. Many studies have found that high levels of anti-SSA/SSB, anticardiolipin, and anti-RNP antibodies predict a higher incidence of PAH [140,142,143]. High levels of anti-U1-RNP antibodies also indicate longer survival periods despite the higher incidence of SLE in patients with PAH [140].

#### 5.7.2. Treatment

Immunosuppressive therapies based on intravenous cyclophosphamide have been successfully used in several studies. As such, intravenous cyclophosphamide and prednisolone reduce the pulmonary artery pressure [144,145]. In addition to immunosuppressive therapies, vasodilators are also effective [146,147]. Multiple studies have demonstrated the efficacy of intravenous epoprostenol (prostacyclin) in normalizing the pulmonary artery pressure in patients with SLE and PAH [148,149]. Bosentan improved the New York Heart Association class and exercise tolerance [150,151]. A case of SLE-associated severe PAH treated with sildenafil was reported in 2003 [152]. A double-blinded study conducted in 2007 showed that sildenafil could improve exercise tolerance, hemodynamic measures, and the World Health Organization functional class in patients with SLE-associated PAH [153].

#### 5.7.3. Sequelae and Outcomes

PAH significantly affects the survival and quality of life in connective tissue diseases, including SLE. It is the third most common cause of death in SLE, following infection and organ failure [154]. As other manifestations of SLE respond well to therapy, PAH is becoming a more important cause of various morbidities and premature deaths [154].

Direct comparisons from the REVEAL registry showed that patients with connective tissue disease-associated PAH had poorer 1-year survival than those with idiopathic PAH (86% vs. 93%). However, patients with SLE had the best 1-year survival rates (94% vs. 82% for scleroderma and 88% for MCTD), although all patients had comparable hemodynamic characteristics at PAH diagnosis [155]. The study cohort was not confined to patients with SLE; however, data from the Korean connective tissue disease (CTD)-related PAH registry, with 174 enrolled patients (61 with SLE, 50 with systemic sclerosis, 10 with mixed CTD, 22 with rheumatoid arthritis (RA), and 31 with other CTDs), demonstrated that the overall 1- and 3-year survival rates of CTD-PAH were 90.7% and 87.3%, respectively [156]. In that study, low DLco, diabetes, and pleural effusion were poor prognostic factors, whereas anti-U1-RNP antibodies seemed to be protective. A recent meta-analysis of six studies encompassing 323 patients with lupus and PAH demonstrated that the pooled 1-, 3-, and 5-year survival rates were 88%, 81%, and 68%, respectively [157].

Additionally, complete autopsy studies of 90 patients diagnosed with SLE (under the ACR diagnostic criteria for SLE) between 1958 and 2006 were performed, and their clinical records were studied [158]. Four of ninety patients showed lesions highly suggestive of PHT (4.4%) [158]. The SLE evolution time ranged from 15 to 23 years. Upon histological examination, they all showed plexiform lesions and organizing microthrombosis, and three of the patients had dilation and hypertrophy of the right ventricle of the heart. At the time of death, three patients had Raynaud’s phenomenon, arthralgia, malar erythema, and hypoxemia, with negative rheumatoid factor and positive ANA [158].

### 5.8. Pulmonary Embolism

#### 5.8.1. Clinical Manifestation

Pulmonary embolism (PE) is a rare but potentially fatal pulmonary complication of SLE that requires careful attention, as most of its causes can be treated. Multiple population studies suggest that patients with SLE are significantly more susceptible to PE and related clinical manifestations, such as deep vein thrombosis, and this association is independent of age, sex, race, and preexisting patient comorbidities [65,159]. The overall prevalence of PE is 1–5% in patients with SLE [159,160]. The risk factors for PE include a high body mass index, fast progression of SLE, hypoalbuminemia, antiphospholipid antibodies, and high doses of glucocorticoids [160].

#### 5.8.2. Treatment

Limited amount of research has been conducted on the treatment of SLE-associated PE; the following recommendations are generally applicable to PE. Early anticoagulation therapy should be used when PE is diagnosed or suspected to reduce its mortality [161]. In patients with massive PE or persistent hypotension (systolic blood pressure < 90 mmHg), the CHEST guidelines recommend thrombolysis as a grade 2 B recommendation [162]. In patients with massive PE, the benefits of thrombolysis usually outweigh the risks unless there is active, uncontrolled bleeding [163].

#### 5.8.3. Sequelae and Outcomes

Complete autopsy studies of 90 patients with SLE diagnosed between 1958 and 2006 and their clinical records were conducted. All patients fulfilled the ACR diagnostic criteria for SLE [8]. Seven patients with pulmonary thromboembolism were detected, four of whom had recurrent PE (4.4%). One case of acute PE was due to catastrophic antiphospholipid syndrome, accompanied by multiple acute arterial and venous thromboses in several organ systems. All patients had active lupus nephropathy at death [158].

Lifelong anticoagulation treatments may be warranted in patients with recurrent thromboembolic disease [164]. In addition, intensive treatment with corticosteroids or immunosuppressive agents may be required when anticoagulation monotherapy fails to control thrombosis [164].

### 5.9. Drug Toxicity and Lung Involvement of SLE

ILD and ALP can be caused by multiple medications, and drug-induced lung diseases should be differentiated from lung involvement in SLE. Several lung diseases are caused by therapeutic agents in patients with underlying diseases such as connective tissue disease and cancer [165]. Since imaging studies can distinguish both conditions, a history of new medications started before evaluating lung involvement in SLE should be considered for diagnosing ILD and ALP [166]. 

**Table 3 jcm-11-06714-t003:** Summary of the types of pulmonary involvement in SLE.

	Clinical Symptom	Treatment	Outcome	Summary	Ref
Lupus pleuritis		Options vary upon the severity of symptoms. NSAIDs can be used for mild pleuritis, and antimalarials may be added [88,89]. Systemic corticosteroid is a treatment choice for severe pleuritis [90]. Aspiration may be needed for massive pleural effusion [87]. Cyclophosphamide, MMF, rituximab can be considered for steroid-resistance or when renal involvement is present [88,92]. The use of IVIG (alone or combined with cyclosporin) has been reported, but further studies are needed [94,95]. Pleurectomy may be needed in rare cases [96].	Prognosis and treatment response are good in general [98]. Severe lung restriction and fibrothorax can occur. Rare complications like fibrothorax can occur, due to misdiagnosis and inadequate treatment [87].	Lupus pleuritis can be treated by immunosuppressive drugs, and has good prognosis in general. Some rare complications can occur, because it can be misdiagnosed initilally.	[87,88,89,90,91,92,93,94,95,96,97,98]
Pleural effusion	The criterion is imaging evidence: US, X-ray, CT, MRI [5]. Pleural effusion in SLE may be due to autoimmune pleuritis, but it’s hard to distinguish it from other causes [6].	It usually shows a rapid response to corticosteroids [99]. Some may have persistent steroid resistant pleural effusion; in such cases tetracycline pleurodesis use and pleurectomy have been reported [100,101,102].	No statistically meaningful studies were found.	Pleural effusion is identified by imaging, and it may be caused by autoimmune pleuritis. It responds well to corticosteroids. There were no studies regarding the outcome of SLE-associated pleural effusion.	[5,6,100,101,102]
Acute lupus pneumonitis		Empirical broad-spectrum antibiotic therapy should be initiated immediately [107]. Aggressive immunotherapy should be given after infectious etiologies are excluded; high dose IV methylprednisolone, oral corticosteroids, IV cyclophosphamide, rituximab, plasma exchnage, IVIG can be considered [91,108,109].	No studies have been reported since 1975, but current treatment has improved the prognosis. Patients with pulmonary infection are at risk of persistent abnormalities of lung function and restrictive lung disease [51,104].	Empirical broad-spectrum antibiotics therapy must be given and aggressive immunosuppressive therapy should be initiated after excluding infections. Studies have not been reported since 1975 about the prognosis, but current options are likely associated with better outcomes.	[51,91,104,107,108,109]
Diffuse alveolar hemorrhage	DAH is common vascular involvement, and it might help to diagnose SLE [1,6].	There is no well established guideline for DAH. High-dose methylprednisolone is often used [110]. Cyclophosaphamide alone and plasma exchange combined with cyclophosphamide or methylpredniosolone are used, without well defined efficacy in SLE-DAH yet [77,119].	SLE-associated pulmonary hemorrhage shows high mortality [132]. Several factors related to high mortality are severe dyspnea, need for mechanical ventilation, renal failure, neuropsychiatric complications, etc [42,111,112,113]. Early and aggressive treatment can increase survival rate [114].	DAH is a quite common and life-threatening condition, showing high mortality rate. High-dose methylprednisolone can be used for treatment.	[1,6,42,76,77,110,111,112,113,114,115,116,117,119]
Interstitial lung disease	SLE-ILD includes NSIP, organizing pneumonia, LIP, follicular bronchiolitis, etc. It typically manifests as an insidious onset of chronic nonproductive cough, dyspnea, decreased exercise tolerance, etc.	Treatment depends on expert opinion [121]. Recent studies suggest corticosteroids alone or with cyclophosaphamide or MMF as an induction therapy, and azathioprine or MMF as a maintenance therapy.	55 SLE-ILD patients showed overall 5yr survival rate of 85.3% [122].Factors predicting worse outcome include current smoking, thrombocytopenia, significant lung fibrosis [122]. Lupus-associated pneumonitis had common progression to respiratory failure and high mortality [111]. It is difficult to estimate the outcome of SLE-ILD.	SLE-ILD includes various types of ILD, and it shows slowly-developing respiratory symptoms. Treatment options are based on expert opinion, but recent algorithms suggest the use of corticosteroids and immunosuppressive agents. Its clinical outcome is hard to estimate accurately.	[103,107,111,118,119,120,121,122,123]
Shrinking lung syndrome	SLS is rare, and it shows a large female predominance [66]. All patients show respiratory symptoms; dyspnea, coughing, pleuritic chest pain. SLS causes progressive decrease in lung volumes on PFT [65,127].	There are no clinical guidelines, Glucocorticoid alone or with immunosuppressive agents are frequently used [66,81,128]. Theophylline, beta-agonists, rituximab alone or with cyclophosphamide and beta-agonist, belimumab can be considered [81,128,129,130,131,132,133,134]	Corticosteroids and immunosuppresion shows a good response. Most will cure from SLS, but some complications such as pneumonia can remain [81,136].	SLS is a rare condition, characterized by progressive decline of lung volumes on PFT. It responds well to corticosteroids and immunosuppresive therapy, but some complications may occur.	[65,66,81,124,125,126,127,128,129,130,131,132,133,134,135,136]
Pulmonary arterial hypertension	It may cause irreversible changes of right ventricle or lung capillaries [137]. Risk factors for PAH include hypertension, high fibrinogen levels, thrombocytopenia, serositis, etc.	Immunosuppresion based on IV cyclophosphamide have been shown to be successful. Other options include IV cyclophosphamide with prednisolone, immunosuppresion in combination with vasodilators, IV epoprostenol, bosentan, and sildenafil [144,145,146,147,148,149].	PAH is the third most common cause of death in SLE [154]. CTD-PAH showed poorer survival than idiopathic PAH [155].	PAH can cause irreversible damage to heart or lung, and it show high mortality rate. Immunosuppresion can be used as a treatment.	[137,138,139,140,141,142,143,144,145,146,147,148,149,150,151,152,153,154,155,156,157,158]
Pulmonary embolism	PE is a rare but life-threatening condition. SLE patients are susceptible to PE. Risk factors include high BMI, fast progression of SLE, hypoalbuminemia, antiphospholipid antibodies, high dose glucocorticoids [70,159,160].	Studies on the treatment of SLE associated PE have rarely been performed. Anticoagulation therapy needs to be commenced [161]. Massive PE and persistent hypotension requires thrombolysis [162].	Life-long anticoagulation may be warranted. Corticosteroids or immunosuppressive agents as a intensive treatment may be required [164].	PE is rare but can be fatal, and SLE patients are at risk of PE. Early anticoagulation should be initiated, and some might need thrombolysis. Some refractory cases may need life-long anticoagulation or intensive treatment.	[8,70,158,159,160,161,162,163,164]

Abbreviations: NSAIDs: non-steroidal anti-inflammatory drugs, MMF: mycophenolate mofetil, IVIG: intravenous immunoglobulin, US: ultrasound, DAH: diffuse alveolar hemorrhage, PAH: pulmonary arterial hypertension, CTD: connective tissue disease, PE: pulmonary embolism, BMI: body mass index, SLS: shrinking lung syndrome, PFT: pulmonary function test, ILD: interstitial lung disease, NSIP: nonspecific interstitial pneumonia, and LIP: lymphocytic interstitial pneumonia.

## 6. Conclusions

A significant number of patients with SLE suffer from pulmonary involvement. Pleurisy, diffuse alveolar hemorrhage, shrinking lung syndrome, interstitial lung disease, and pulmonary arterial hypertension are the main manifestations of lung diseases. Some patients experience critical complications, such as pulmonary hemorrhage. Therefore, the assessment and treatment of lung involvement in patients with SLE should be performed promptly. CXR, HRCT, and PFT are recommended as diagnostic work-up. Clinicians and patients are less aware of the effects of SLE on their respiratory system. Moreover, exact diagnostic criteria for lung involvement in SLE remain elusive. Thus, more attention should be paid to the active surveillance and management of pulmonary manifestations of SLE.

## Figures and Tables

**Figure 1 jcm-11-06714-f001:**
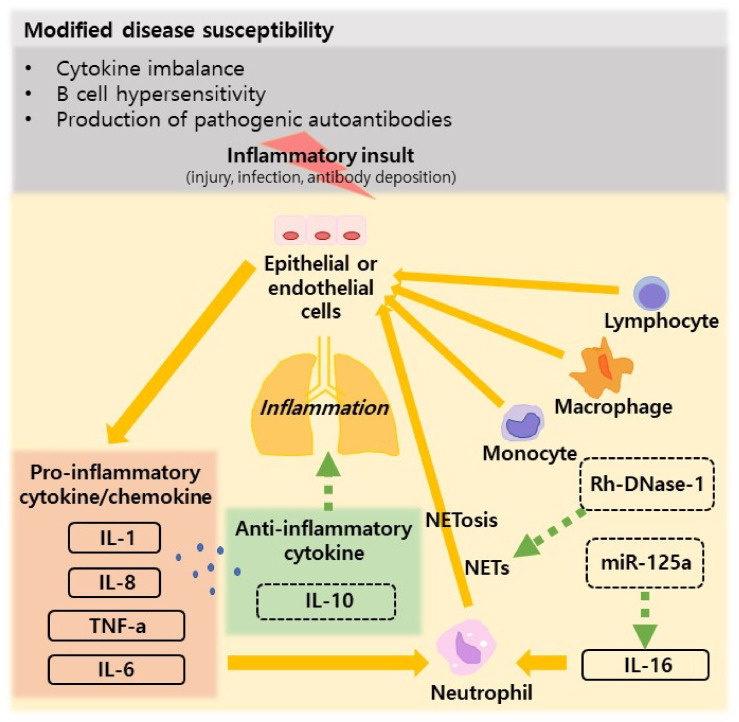
Pathogenesis of lung involvement in SLE [50,51,59,60].

**Figure 2 jcm-11-06714-f002:**
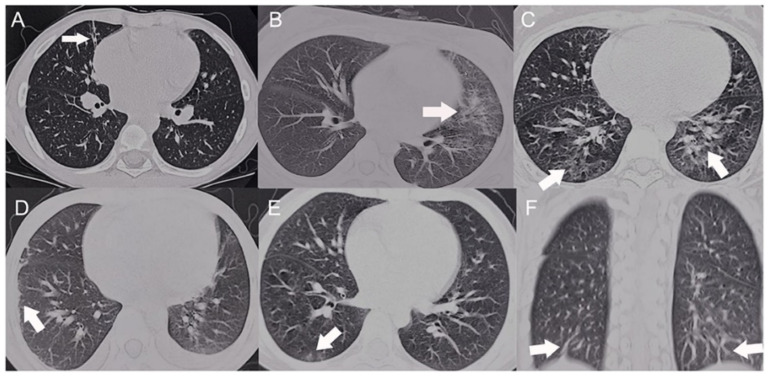
HRCT scans of the SLE patients with pulmonary involvement. Arrow indicate the shape of lung lesion of HRCT (**A**) Fibrotic streak. (**B**) Reticular pattern. (**C**) Mosaic perfusion. (**D**) Pleural thickening. (**E**) Ground glass opacity. (**F**) Subpleural interlobular septal thickening (The CC-BY Creative Commons attribution license, Dai G, Li L, Wang T, Jiang W, Ma J, Yan Y, Chen Z. Pulmonary Involvement in Children With Systemic Lupus Erythematosus. Front Pediatr. 2021 Feb 2; 8:617137. doi: 10.3389/fped.2020.617137. PMID: 33604317; PMCID: PMC7884320) [69].

**Table 1 jcm-11-06714-t001:** Studies on the pathogenesis of lupus-associated lung involvement.

Published Year	Study Type	Key Finding	Comments
Nielepkowicz- Goździńska A et al. (2013) [57]	In vivo	Exhaled cytokines in SLE with lung involvement	The mean IL-6 and IL-10 concentrations in the BALF and the IL-10 concentration in the EBC were higher in patients with SLE compared with healthy controls. Study showed that IL-6 and IL-10 are involved in the pathogenesis of SLE and it is likely that IL-10 protects against pulmonary manifestations of SLE.
Zhuang H et al. (2017) [58]	In vivo	Pathogenesis of diffuse alveolar hemorrhage in murine lupus	The pathogenesis of DAH involves opsonization of dead cells by natural IgM and complement followed by complement receptor-mediated lung inflammation. The disease is macrophage dependent, and IL-10 is protective. Complement inhibition and/or macrophage-targeted therapies may reduce mortality in lupus-associated DAH.
Smith S et al. (2018) [59]	In vivo/ In vitro	IL-16/miR-125a axis controls neutrophil recruitment in pristane-induced lung inflammation	Neutrophil infiltration was markedly reduced in the peritoneal lavage of pristane-treated IL-16-deficient mice and elevated following administration of IL-16. miR-125a mimic reduced pristane-induced IL-16 expression and neutrophil recruitment and abrogated the induced lung pathology. To sum up, IL-16 acts directly on the pulmonary epithelium and markedly enhances neutrophil chemoattractant expression both in vitro and in vivo, while the miR-125a mimic can prevent this.
Jarrot P-A et al. (2019) [60]	In vivo	NETs are associated with the pathogenesis of diffuse alveolar hemorrhage in murine lupus	PMNs and NETs play an important pathogenic role in lung injury during pristane-induced DAH. Targeting NETs with Rh-DNase-1 inhalations could be an interesting adjuvant therapy in human DAH.

Abbreviations: SLE: systemic lupus erythematosus, BALF: bronchoalveolar lavage fluid, EBC: exhaled breath condensate, IgM: immunoglobulin M, DAH: diffuse alveolar hemorrhage, PMN: polymorphonuclear neutrophil, NETs: neutrophil extracellular traps, miR: microRNAs, and Rh-DNase: recombinant human deoxyribonuclease.

**Table 2 jcm-11-06714-t002:** Summary of the characteristics of the diagnostic tools to diagnose pulmonary involvement in SLE.

	Key Finding	Comments	Ref
CXR	CXR findings vary upon diseases, and usually correlate with clinical symptoms [35].	CXR is the most basic imaging method, and its sensitivity is low.	[35,61,62,63,64,65,66]
CT	70% of the patients showed abnormal findings such as interlobular septa thickening, parenchymal bands, etc [67].	CT imaging is useful to make an early diagnosis of pulmonary involvement, and it provides precise expectation of pulmonary function [64].	[67,68,70]
PFT	DLco shows the most prominent abnormality among the spirometric values [71]	PFT provides an option to early detect of pulmonary involvement in SLE patients, but it lacks sensitivity [71,72].	[71,72]
Biopsy	Pleural involvement shows results such as fibrosis, lymphocytic/plasma cell infiltration, etc. ALP show alveolar wall damage and necrosis, etc. Bland hemorrhage is predominantly found in DAH [45,47,73].	Biopsy is rarely used in the diagnostic work-up; it is used only when the diagnosis is unclear and other findings are nonspecific, and it is avoided due to postoperative complications [45].	[45,47,73]
BAL	It shows increased cellularity in ALP, lymphocytic/neutrophilic predominance in ILD, predominance of macrophage in SLE, and the presence of hemosiderophage in DAH [35,51,75,76,77].	BAL is mainly performed to exclude other causes such as infections [35].	[35,51,75,76,77]
FDG-PET	SLE disease activity and immune/inflammatory status is correlated with high FDG uptake PAH in patients with SLE [79].	It can be the potential marker for intrapulmonary disease activity in SLE-PAH patients [79].	[79]
Dynamic MRI	Dynamic MRI sequences on forced voluntary ventilation can confirm physiologic movements of both diaphragms and all auxiliary respiratory muscles	Magnetic resonance imaging-confirmed pleuritis in systemic lupus erythematosus-associated shrinking lung syndrome	[80,81]

Abbreviations: CXR: Chest X-ray, CT: computed tomography, SLE: systemic lupus erythematosus, PFT: pulmonary function test, DLco: diffusing capacity for carbon monoxide, ALP: acute lupus pneumonitis, DAH: diffuse alveolar hemorrhage, BAL: bronchoalveolar lavage, ILD: interstitial lung disease, FDG: fluorodeoxyglucose, PET: positron emission tomography, PAH: pulmonary arterial hypertension, and MRI: Magnetic Resonance Imaging.

## Data Availability

Data are available from the corresponding author on a reasonable request.

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
