# Peer review of "Systemic Lupus Erythematosus and Lung Involvement: A Comprehensive Review"

_jcm, 2022, doi:10.3390/jcm11226714_

Round 1

Reviewer 1 Report (Previous Reviewer 2)

The authors have responded appropriately to all the advice and minor revisions

Author Response

Thanks for your recommendation

Reviewer 2 Report (New Reviewer)

Dear Authors! Very intersting and useful manuscript from praticing point of view. Lung involvement is one of the severest manifestation of SLE related with life expactancy.

I have several suggestions:

some SLE patients develop acute respiratory distress syndrome. In you manuscript is the same as acute pulmonitits? Please discuss about the role of macrophage activation syndrome in some form of lung involvement in SLE as well as APL syndrome.

Please provide literature data about predictors of lung involvement (especially silent as PAH) to early identifications of some form of lung involvement

English editing srtonly required

Author Response

Thanks for your recommendation.

I add response to your comments

This manuscript is a resubmission of an earlier submission. The following is a list of the peer review reports and author responses from that submission.

Round 1

Reviewer 1 Report

Shin et al aimed to give an overview of pleuro-pulmonary involvement in SLE. This review is well detailed. However, some statements could be justified with more references. 

Comments:

What is the difference between pleural effusion and pleuritis for the authors? I think it could be classified in the same group in the introduction.

What is the place for 99m Tc-HMPAO perfusion scan and FDG-PET? Dynamic MRI of the diaphragm should be added for the diagnosis of shrinking lung syndrome.

For incidence and prevalence data the authors may refer to several to studies to give a range rather than giving one data for example the date about acute lupus pneumonitis relies on one reference only (ref [35]) which is quite old (2005). 

Page 4 lines 168-169: “It is reported that SLE patients with lung involvement have increased levels 168 of systemic pro – inflammatory cytokines.” Please precise in the text if such levels are measured / observed in blood or in lungs.

Page 6 line 202: “In ALP, CXR demonstrates the evidence of pulmonary hypertension”: a X Ray cannot show pulmonary hypertension; it can show signs suggestive of pulmonary hypertension but this exam is not sufficient to assert the diagnosis. Same remark for lines 208-209: “In the case of DAH, the radiograph might show diffuse hemorrhage”: replace by signs suggestive of DAH and detail those signs.

Page 7 line 278: for BAL, it is false to declare that “BAL is mainly used to exclude other causes.” This exam is indeed important to rule out an infection but it is the gold standard to confirm the diagnosis of intra alveolar hemorrhage. 

Page 7 line 287: “And it was different from SLE without pulmonary symptoms, which showed the predominance of macrophages.” A sentence cannont begin with “And”.

Page 7 line 286: “4.6.99. m Tc-HMPAO Perfusion scan”. For this paragraph, the authors refer to one sole research study. I would not consider this exam to be routinely used in the diagnosis of lung involvement related to SLE.

Page 8 line 304: “Therefore, the definition, which is precise and covers various pulmonary diseases by using many tools needs to be established.” This sentence is not clear as the rest of the paragraph “summary” of the diagnostic tools. The aim of a review is to summarize the literature but also to analyze and interpret it as experts of the disease. In this setting, the authors should state that pulmonary screening in SLE should include chest CT and LFTs that are useful and complementary exams, not too invasive and easily available and that are able to detect early manifestations sometimes before the occurrence of symptoms. In this section, the authors, should also write a paragraph on MRI of the diaphragm to diagnose shrinking lung syndrome.

Page 10 line 307: “Unfortunately, there is no clear definition of pulmonary involvement in SLE now. So, the definition of pulmonary involvement in SLE needs to be clarified.” I do not understand this assertion. Lung involvements in SLE are well known and as the authors write in the introduction, those involvements are multiple with pleural effusion / pleuritis, shrinking lung syndrome, acute pneumonitis, DAH, chronic ILD and pulmonary hypertension.

Page 10 line 346: “In the absence of a pleural effusion on the chest X-ray, it is often difficult to diagnose pleuritis. 347 Pleural friction rub may facilitate the diagnosis of pleuritis”. Ok but if chest X ray is not conclusive and that pleuritis is suspected, pleural US or chest CT can be performed to look for pleuritis.

Page 11 line 409: “As lupus pleuritis is the most common pulmonary complication of SLE, prognosis and treatment response are good. I do not catch the link between the 2 parts of this sentence: why should the fact that pleuritis is the most common complication of SLE should explain the good prognosis and treatment response ?

Page 12 line 471: “Definition should cover various types of pulmonary involvement, range 471 from mild pleural effusion to severe pulmonary hemorrhage. Parenchymal 472 involvement in SLE includes ILD and ALP. [6] ILD is uncommon in SLE and 473 exhibits a slow progress, while ALP has a high mortality, and might be hard 474 to distinguish from infection and acute respiratory distress syndrome. [6]” This part is out of the subject of the paragraph DAH and should be deleted.

Page 13 line 515: “In SLE patients with PAH, low values of ratio between tricuspid annular plane systolic excursion (TAPSE) and pulmonary artery systolic pressure (PASP) predicted poor prognosis, as well as low 6-minute walk distance (6MWD).” Why is this sentence placed in the DAH paragraph ?

Reviewer 2 Report

The article clearly and exhaustively covers the pleuropulmonary manifestations of SLE. I agree with the authors that this article "closes a gap" of the literature on pleuropulmonary involvement in SLE and will be of great clinical support for physicians and teams caring for patients with lupus.

The historical reminder is pleasant and the article provides some points of consideration for future research.

As a pediatric radiologist, I can say that the shrinking lung syndrome is quite unknown among my peers and probably under-diagnosed in teenagers with SLE. A more detailed description of the radiological aspect and a reference on the contribution of dynamic diaphragmatic imaging (ultrasound or MRI) may be of interest.

The section on the clinical manifestations of DAH is not clear, it needs to be better described and expanded. A short specific pathophysiological description might be of interest.

Does the format of the journal allow for imaging illustrations? If so, it would be really interesting to add examples of HRCT images of ALP, DAH and NSIP, at least in additionnal content. If not, a detailed description of the HRCT features is required in my opinion, and will increase the practical and clinical value of the article.

The differential diagnosis between ILD, ALP and drug toxicities should be emphasized, as it is not negligible in systemic autoimmune diseases. For example, the authors could add a small paragraph mentioning the usual lupus therapies that may induce pulmonary toxicity and the need to perform chest imaging if pulmonary symptoms appear after the introduction of one of these therapies. 
